# Gender Differences in Performing an Overhead Drilling Task Using an Exoskeleton—A Cross-Sectional Study

**DOI:** 10.3390/biomimetics9100601

**Published:** 2024-10-07

**Authors:** Bettina Wollesen, Julia Gräf, Sander De Bock, Eligia Alfio, María Alejandra Díaz, Kevin De Pauw

**Affiliations:** 1Department of Human Movement Science, Universität Hamburg, 20148 Hamburg, Germany; julia.graef@uni-hamburg.de; 2Institute of Movement Therapy and Movement-Oriented Prevention and Rehabilitation, German Sports University Cologne, 50933 Cologne, Germany; 3Human Physiology and Sports Physiotherapy Research Group, Vrije Universiteit Brussel, 1050 Brussels, Belgium; sander.de.bock@vub.be (S.D.B.); eligia.alfio@vub.be (E.A.); ma.diaz@vub.be (M.A.D.); kevin.de.pauw@vub.be (K.D.P.); 4Human Robotics Research Center, 1050 Brussels, Belgium

**Keywords:** supportive systems, gender comparison, over-shoulder working task

## Abstract

(1) Exoskeletons offer potential benefits for overhead working tasks, but gender effects or differences are unclear. This study aimed to compare the performance as well as subjective body strain and comfort of men and women using an upper-body exoskeleton. (2) n = 20 female and n = 16 male participants performed an overhead drilling task with and without a passive upper-body exoskeleton in a randomized cross-over study. The task performance of different movement phases, perceived exertion, and ease of use were measured to compare gender differences. One- and two-way analyses were used to compare genders in the different conditions. The body mass index (BMI) was included as a covariate. (3) Gender differences in task performance were found for error integrals (*p* < 0.001) with higher values in male participants. Moreover, there was a significant interaction effect for gender x exoskeleton use. While females showed performance decrements in aiming with exoskeleton use, the males’ performance increased (*p* = 0.025). No other gender differences were observed. (4) Gender differences in task performance using an upper-body industrial exoskeleton were less detectable than expected, indicating that body composition and anthropometrics might be valuable indicators for performance including assisting devices. Moreover, future studies should also integrate the examination of muscle activity to gain more insights into potential gender movement control patterns.

## 1. Introduction

Exoskeletons have emerged as innovative tools to augment human physical capabilities, offering promising applications across various fields, including rehabilitation and industrial work [1]. Among these, upper-body shoulder exoskeletons are designed to assist with tasks that involve repetitive lifting, holding, or overhead activities, aiming to reduce the physical strain and the risk of musculoskeletal disorders [2,3].

Despite the growing body of research on exoskeleton efficacy, there remains a critical gap in understanding how these devices affect diverse user groups, particularly between genders. Men and women often exhibit physiological and biomechanical differences, which can influence the performance and effectiveness of assistive devices like exoskeletons [4]. Factors such as muscle strength or range of motion according to body dimensions vary between genders and may lead to differing interactions with the exoskeleton [5].

Men generally have higher absolute muscle strength, particularly in the upper body, compared to women [6,7,8]. This can affect how men utilize the exoskeleton’s support in tasks requiring high-force output [9]. Women may have comparable muscle endurance, but their lower muscle mass can influence how the exoskeleton’s torque distribution and load bearing are perceived, potentially affecting long-term use and fatigue rates [8,10,11].

Men and women differ in body proportions, including differences in shoulder width, arm length, and torso dimensions. Dissimilarities in body composition, such as a higher percentage of body fat in women and greater muscle mass in men [12], can affect the anthropometric fit and therefore the functionality of exoskeletons. These differences may influence how body weight and the additional weight of the exoskeleton are distributed and how the exoskeleton stabilizes the body during use. In addition, women might adopt different postural adjustments due to their wider pelvis and different center of gravity, which can further impact the exoskeleton’s fit and function [13].

Moreover, women tend to have a greater range of motion in their shoulder joints compared to men [14], which could affect how they interact with the exoskeleton. This increased flexibility might allow for a greater range of overhead motion but could also lead to other requirements from the exoskeleton regarding stabilization and support [15]. Women may also adopt distinct postural adjustments and kinematic strategies when using an exoskeleton for overhead tasks and might therefore show distinct movement patterns compared to men, which could be influenced by the exoskeleton’s assistance [16]. Additionally, women might excel in tasks requiring fine motor control and precision [17], which could interact uniquely with the exoskeleton’s assistance. This might result in different task performance metrics such as speed, accuracy, and coordination compared to men [16,17,18,19].

The study by Gräf et al. [19] has shown that when wearing an upper-body exoskeleton while executing overhead tasks, errors in cognitive tasks, especially under fatigue conditions, were significantly reduced. Furthermore, while muscle fatigue led to a general decrease in movement time to finish a task, it did not alter the overall speed–accuracy trade-off, indicating the motor system’s ability to compensate and adapt.

All these variations can impact the fit and comfort of the exoskeleton, potentially affecting performance and ease of use. The poor fit of the exoskeleton caused by these differences can lead to discomfort or suboptimal support during tasks, influencing motor performance [20]. Therefore, women might report higher levels of discomfort or perceived exertion when using an exoskeleton designed with a male anthropometry in mind. Females might experience discomfort or suboptimal support if the exoskeleton is not designed to accommodate their body proportions [21], potentially affecting precision and work productivity [22,23,24].

Understanding the gender differences in wearing an upper-body exoskeleton is crucial for the development of designs that are inclusive and effective for all users. Addressing these differences involves considering physiological, biomechanical, and ergonomic factors to optimize fit, comfort, and performance.

While general trends suggest that motor precision could differ between males and females when using an exoskeleton, the extent and nature of these differences are highly dependent on individual physiological and biomechanical factors, as well as the design and adjustability of the exoskeleton itself. Tailoring exoskeletons to accommodate gender-specific needs can help optimize assistance, work productivity, and precision motor precision for both males and females.

This study aims to compare the effects of wearing an upper-body shoulder exoskeleton between women and men by assessing task performance of an overhead nailing task including perceived exertion and comfort. We seek to determine if there are significant gender-based differences in the use of the device.

Since the exoskeleton was designed and constructed for male users, we hypothesize that women will perform worse during the drilling task and have a higher rating of perceived exertion and lower usability score when using the exoskeleton in comparison to male users.

The findings of this study could inform the design and development of more inclusive exoskeletons, ensuring optimal performance and safety for all users, regardless of gender.

## 2. Materials and Methods

### 2.1. Study Design

This study followed a 2 × 2 randomized balanced cross-over study design (cf., Figure 1).

### 2.2. Participants

#### 2.2.1. Sample Size Calculation

To compare male and female participants with and without exoskeleton support, a power calculation with G*Power revealed a total sample size of 41 subjects (ANOVA; fixed effects, main effects and interactions, ß-Power of 0.8, estimated effect size of 0.45, and alpha of 0.05).

#### 2.2.2. Inclusion and Exclusion Criteria

Biological females and males aged 18 and older, right-handed, and without a history or current manifestation of musculoskeletal disorders or injuries in the upper extremity within the past 12 months were included.

Participants were excluded if they had any injuries or health-related complaints. Only right-handed participants were included.

#### 2.2.3. Participant Description

Participants were recruited on or in the surroundings of the VUB campus in two rounds. First, a number of n = 16 males were examined. Afterwards, a total number of n = 21 females were incorporated into the experiment. n = 1 female needed to be excluded due to poor data quality.

The mean age of the sample was 31.2 ± 14.5 years for the females and 31.9 ± 9.1 years for the males.

The anthropometrics naturally differed between female and male participants with higher values for males (cf., Table 1). As the BMI also differed significantly, the performance data had to be controlled for BMI.

#### 2.2.4. Restrictions and Prohibitions for the Subjects

Subjects were asked not to drink caffeine-containing beverages or alcohol and to refrain from vigorous training 24 h before testing to avoid confounding influences on cognitive and physical performance.

### 2.3. Measurements

#### 2.3.1. Anthropometric Data

To unify the working situation, the height of the working position needed to be adjusted to body height including the arm length and the hand/elbow position during the drilling task. Therefore, next to common anthropometrics like body height and body mass, the fingertip height and the fingertip height with the elbow at shoulder height with 90-degree flexion were also measured.

#### 2.3.2. Drilling Performance

To quantify the working performance and facilitate segmentation of the acquired signals, force sensors and accelerometers were attached to the overhead working set-up and the weights that would be lifted. To determine the proper overhead height, the method of Sood et al. [25] was used (hand height with the shoulder and elbow in a 90-degree angle + 0.4 × (hand height with the arm in full extension—hand height with the shoulder and elbow in a 90-degree angle). To evaluate the precision performance in overhead work, a custom task with high to moderate test–retest reliability (adapted from Kim et al. [26]) was developed. In this task, participants used an electric screwdriver (Black & Decker, New Britain, CT, USA, 1.14 kg) to tighten 20 pre-inserted bolts in an aluminum plate positioned at overhead height. We determined that a task was successfully completed until all 20 bolts were tightened (mean task duration). The bolts were partially screwed into the plate using self-fixing nuts, standardizing the distance between the bolt heads and the plate. A plexiglass mask was placed over the bolts to interfere with the screwdriver tip if the screwing trajectory deviated by more than ±2° from perpendicular to the plate [26]. Participants pressed a push button at pelvic crest height, tightened a bolt at overhead height, and pressed the button again (cf., Figure 2), repeating this process for all bolts. They were instructed to perform the task quickly and accurately, minimizing errors. Sensorized aluminum and plexiglass plates tracked the contact between the screwdriver bit and the bolt, as well as screwing errors. Pressing the push button marked the start of each movement.

During the overhead drilling task, the exoskeleton’s encoder data and load cell data of the sensorized overhead work bench were acquired at a frequency of 1000 Hz, using Beckhoff components (Beckhoff, Verl, Germany) interconnected via an Ethercat communication protocol. Pressing the push button at hip-level indicated movement initiation. The load cells connected to the bolts allowed tracking when the screwdriver bit was touching a bolt. The encoder data were used to differentiate between lifting the arm from hip level towards the bolts, and the slower aiming process before the bolts were reached. These data allowed all acquired data to be segmented into a lifting and aiming phase, and operative phase. In the latter phase, the bolts were tightened. Data from the load cells connected to the plexiglass surrounding the bolts documented the error. In order to encompass the duration of the error, the error score was computed as the time-integrated output from the force sensors connected to the plexiglass plates.

The Exo4Work’s right shoulder joint, fitted with an RM22 Rotary Magnetic Encoder, tracked shoulder elevation angle (1 kHz; RLS, Komenda, Slovenia). All measurements were recorded on the right side, so tasks were performed right-handed, with the left hand at rest.

#### 2.3.3. Perceived Exertion BORG Scale

For the male participants in this study, the 10-point Borg scale was used [27,28].

For the second part of this study, including female participants, the subjective assessment of perceived exertion was carried out using the 100-point Borg scale [27,28,29,30], which is more precise in accounting for the RPE. This scale allows participants to subjectively assess the intensity of exertion after each drilling trial, with 0 representing no exertion and 100 indicating maximal effort. The assessment refers to the overall perceived exertion and specific body regions, such as the neck, shoulders, arms, upper and lower back, and legs. A chart was shown for this purpose. The rating of the perceived exertion was expressed verbally by the participants.

The familiarization to the RPE scales was performed by verbal instructions in the initial session. Familiarizing participants with the scale involved presenting the scale and asking for their ratings during the experimental trial. This process helped participants become accustomed to the task and measurement tools being used.

To compare both CR scales, we downscaled the female 100 CR scale to the 10-point scale.

#### 2.3.4. System Usability Scale (SUS)

The System Usability Scale (SUS) provides a reliable tool for measuring usability, comprising ten questions with response options ranging from “Strongly Agree” to “Strongly Disagree” [31]. This tool facilitates the assessment of diverse products and services, encompassing hardware, software, mobile devices, websites, and applications. 

#### 2.3.5. Passive Shoulder Exoskeleton Characteristics

The upper-body exoskeleton (Exo4Work) used in this study was developed by the research group of the Vrije Universiteit Brussel and KU Leuven. This industrial exoskeleton weighs 3.8 kg and features 6 degrees of freedom to ensure compatibility with the glenohumeral joint. The support offered by the exoskeleton can be modulated by adjusting the pretension of the spring, an element of the passive remote actuation mechanism. The exoskeleton can provide upper-body assistance of 0.5 Nm within a shoulder flexion range of 0–35° with the support peaking at 3 Nm at 105°. The set-up including pretension of the spring was standardized and kept the same for each subject. The exoskeleton can be worn on the body like a backpack with a hip belt, which ensures the transfer of the weight from the back to the legs. The device was attached to the upper arms with Velcro straps, similar to commercially available devices. For more information about the exoskeleton (kinematics, torque distribution, and hysteresis), readers should refer to Rossini et al., 2021 [32].

### 2.4. Procedures

All tests took place at the laboratory of the Human Physiology and Sports Physiotherapy Research Group (MFYS, U-Residence, campus Etterbeek). This study was conducted by the standards of the Declaration of Helsinki and the local medical ethics committee (Vrije Universiteit Brussel and Universitair Ziekenhuis Brussel, B.U.N.: 143201941463). Upon initial arrival, the subject was informed about the protocol and signed an informed consent.

The experiment consisted of a total of three laboratory visits in which general participant characteristics (e.g., body height and weight, measurements of body parts) were initially recorded (visit 1) and participants were introduced to the experimental protocol, the laboratory environment, and the Exo4Work exoskeleton (familiarization)—a passive upper-body exoskeleton (PSE). Over the following two visits, the participants completed the experimental protocol, which followed a 2 × 2 randomized counterbalanced cross-over approach (with and without an exoskeleton). Each trial lasted approximately 1.5 h. In between the first and the second visit, at least 48 h was scheduled. In between the second and the third laboratory visit, 6 to 9 days was foreseen.

#### Familiarization

The first visit to the lab included familiarization with the study protocol and the exoskeleton to get to know the routine and reduce learning effects throughout the experimental trials. This involved the execution of the overhead precision task with the Exo4Work with a duration up to, 12 times including with a 3 min break in between each repetition. The session was terminated when the participants became physically very tired.

After completing the task, participants rated their physical fatigue on the 10-point or 100-point Borg Scale and completed the SUS questionnaire to assess system usability.

### 2.5. Statistics

To analyze the data, one-way ANOVA was performed to compare demographic variables (body weight and height, BMI, fingertip height, and fingertip height with elbow at 90 degrees) to detect gender differences. Moreover, a two-way ANOVA was conducted to compare male and female data in terms of performance, perceived exertion, and scoring of the SUS. The BMI was included as a covariate to control for its potential influence on the outcomes. A Bonferroni correction was applied to adjust the alpha level for multiple comparisons. For all statistical tests, the significance level was set at α = 0.05. Statistical analysis was conducted with SPSS 29.0.

## 3. Results

### 3.1. Drilling Performance

As shown in Table 2, there were no gender differences for the mean task duration; however, the task duration changed with exoskeleton use and increased for females and decreased for males (F(1,33) = 0.954; *p* = 0.327; eta^2^ = 0.030). Nevertheless, this difference failed to be significant if the data were controlled for BMI (cf., Table 2 and Figure 3).

However, there were gender differences for the mean error integral for aiming (F(1,33) = 25.153; *p* < 0.001; eta^2^ = 0.440), even when controlling for BMI (F(1,33) = 5.509; *p* < 0.001; eta^2^ = 0.147). Furthermore, there were differences in the mean error integral for drilling with overall higher errors for the males (F(1,33) = 16.766; *p* < 0.001; eta^2^ = 0.344). Moreover, for the mean errors of aiming, the error rate increased significantly with exoskeleton use for the aiming phase, while the errors of males decreased (cf., Table 2 and Figure 4).

### 3.2. Perceived Exertion

The RPE analysis did not show any differences in the main effect of condition (with and without exoskeleton) and gender. Both genders had a score of 4.0 ± 1 for working without an exoskeleton and 3.8 ± 1.5 for working with an exoskeleton (F(1,33) = 0.472; *p* = 0.397, eta^2^ = 0.014). There was a between-subject effect for the BMI (F = 11.985; *p* = 0.033; eta^2^ = 0.131).

### 3.3. System Usability

The evaluation of the system usability did not lead to significant gender differences for the SUS score (female score: 31.3 ± 16.7; male score 22.8 ± 11.1; F(1,30) = 2.825; *p* = 0.103; eta^2^ = 0.086).

## 4. Discussion

Motor precision, or the ability to execute movements with accuracy and control, varies between males and females when using occupational exoskeletons, influenced by a multitude of factors. While specific studies on motor precision differences in gender, regarding exoskeleton use, are limited, general trends in motor control and biomechanical differences may provide valuable insights. Females tend to have superior fine motor control and precision in certain tasks including overhead work, which could influence how they use an exoskeleton for precision-dependent activities. Males have greater gross motor control, which could impact tasks requiring strength and broad movements more than those requiring fine precision. The movement patterns are determined by the muscle activation patterns [17,33]. These patterns can influence how the exoskeleton assists with precision tasks, potentially making it more effective for one gender over the other depending on the design. Therefore, this study aimed to compare the performance of an overhead precision drilling task between males and females with and without exoskeleton use. We hypothesized that females would exhibit slower task performance, especially when using the exoskeleton, potentially attributed to greater discomfort.

### 4.1. Performance

Surprisingly, the results of the mean duration of the task performance during the drilling task showed no significant differences for females and males. We supposed this in line with previous research, suggesting that females generally take more time in tasks requiring fine motor control due to differences in muscle strength and endurance [34]. However, controlling the analysis by the covariate BMI, these expected differences were not observed.

Next to this, for the main duration of the task performance as well as for the drilling part, the values worsened for the females and improved for the males (cf., Table 2), but this difference failed to be significant (*p* = 0.071). These findings were also not expected. We assumed that the use of an upper-body exoskeleton might mitigate potential differences, reducing the task duration for both genders, and that exoskeletons can be effective in equalizing performance between genders by compensating for physiological differences, particularly in tasks requiring sustained muscle engagement [35]. However, the exoskeleton support did not alter drilling times across genders, implying that drilling efficiency is less influenced by exoskeleton support and more dependent on individual skill or strength, which aligns with previous findings that highlight gender differences in tasks requiring upper-body strength [6]. Moreover, women typically have a greater range of motion in the shoulder joints, which can affect how they interact with the exoskeleton [33]. This increased flexibility might require adjustments in the exoskeleton’s design to accommodate a wider range of movement without causing discomfort or reducing effectiveness. It has to be noted that these results were observed including the BMI as a control variable in the measurements following the results of a systematic review by Cote [17]. As performance differences in movement patterns might also be explained by strength and endurance, and not by gender itself, these aspects should be integrated into future study designs to avoid a potential bias in the data [17].

Regarding the errors, the mean error integral for aiming and drilling was higher in males, which might be attributed to a more aggressive or less precise approach often observed in male participants in similar tasks [36]. Men and women often exhibit different movement patterns and strategies for performing tasks. Men rely more on strength, while women might utilize more coordinated and precise movements. These differences can influence how effectively each gender uses the exoskeleton to enhance performance and reduce strain. Interestingly, the use of an exoskeleton increased the error rate for aiming in females, while it decreased for males. This could indicate that while the exoskeleton provides stabilization that benefitted males, it could have imposed constraints on females, potentially due to differences in body mechanics and anthropometrics or how each gender interacts with assistive devices [37,38]. The observed increased errors for females suggest that exoskeletons might need to be tailored differently for each gender to optimize performance, aligning with previous research indicating that assistive technologies often need gender-specific adjustments [39].

### 4.2. Perceived Exertion

Contrary to the performance metrics, the analysis of perceived exertion (RPE) revealed no significant differences between genders, either with or without the exoskeleton. Both genders reported similar exertion levels, suggesting that the exoskeleton’s support did not differentially impact the subjective workload of the task. This finding contrasts with studies that have reported higher perceived exertion in females during physically demanding tasks, possibly due to different muscle fatigue rates [34,40]. However, the absence of a significant difference in this study may be due to the relatively moderate intensity of the task.

The notable effect of BMI on perceived exertion (with a higher BMI associated with increased exertion) underscores the importance of considering individual physical characteristics in the design and use of exoskeletons. This finding aligns with the literature that highlights the influence of body composition on task performance and perceived exertion, especially in tasks requiring sustained upper-body engagement [35].

### 4.3. System Usability

The evaluation of system usability while executing tasks with the upper-body exoskeleton, using the System Usability Scale (SUS), revealed no significant gender differences, though the scores were marginally higher for females. Despite the observed gender-specific differences in performance, both genders reported similar levels of usability for the exoskeleton, suggesting that the device is generally well suited for both males and females in terms of usability. The lack of significant differences in SUS scores could imply that the perceived usability of the exoskeleton was not strongly affected by the actual performance outcomes. This is consistent with studies suggesting that usability perceptions are often more influenced by the subjective experience of ease of use rather than objective performance metrics [31,41,42]. Moreover, there is some evidence that exoskeleton acceptance combined with training on self-efficacy to use the device was associated with the willingness to continue using the device [43].

The slightly higher SUS scores for females, though not statistically significant, might indicate a tendency for females to rate the usability of assistive devices more positively. However, given the small effect size, this observation warrants further investigation with larger sample sizes or diverse usability measures and setting options to determine if there is a meaningful difference in usability perceptions between genders. This could provide a more comprehensive understanding of any subtle gender-related perceptions or preferences, e.g., adaptations to physical requirements, especially for female users regarding system usability.

### 4.4. Strength and Limitations

One strength of this study is the age- and gender-balanced cohort of this study, securing comparability of the task performance. Moreover, the experiments were conducted in a highly standardized environment mimicking a real industrial environment and controlling for external factors that could have influenced performance.

However, there are also some potential limitations. We found a significant influence of BMI on the performance data. We suppose that the individual upper-extremity maximal strength might have gained more insights into the performance differences between the participants. Therefore, we suggest controlling for maximum strength and strength endurance for future gender comparisons.

Next to this aspect, the experimental set-up was designed and restricted for right-handed persons. Therefore, the results do not represent potential differences in the performance of left-handed individuals and could not be generalized for left-handed persons.

We have to address the fact that the individuals´ shoulder angle might differ from the exoskeleton shoulder support angle. We only measured the angle of the exoskeleton shoulder joint.

Moreover, the participants were not recruited in industrial settings and were therefore not familiar with the task performance. On the other hand, as our analysis integrated in-between subject comparisons for the task conditions with and without exoskeletons as well as a randomized order within the experiment, this disadvantage of the participant group might be relatively small.

Additionally, the use of a specific drilling platform in this study might limit replication in future studies. Nevertheless, future studies should also use real-world industrial scenarios to confirm whether there are differences in lab versus real-world studies.

In summary, further research is needed to develop exoskeletons that can be easily adjusted to accommodate a wide range of body types and movement patterns, ensuring that both men and women can benefit equally from these advanced assistive technologies.

## 5. Conclusions

The analysis showed significant gender-related variations in task performance, particularly in the error rates during aiming and drilling.

The effectiveness of the exoskeleton in assisting with specific tasks vary between genders due to differences in strength, endurance, and movement patterns. Women might benefit more from exoskeletons that enhance precision and fine motor control, while men might see greater benefits from those that augment brute strength and load-bearing capacity. However, some of these effects are reduced if the performance data are controlled for BMI.

Nevertheless, these differences did not translate into changes in perceived exertion or system usability, indicating that while performance metrics may vary, the subjective experience of using the exoskeleton remains similar across genders.

Ensuring that the exoskeleton is anthropometrically designed to fit both male and female body types is crucial. An ill-fitting exoskeleton can lead to decreased performance, discomfort, and even injury. Adjustability in key areas, such as shoulder straps, torso supports, and arm cuffs, can help accommodate different body shapes and sizes.

## Figures and Tables

**Figure 1 biomimetics-09-00601-f001:**
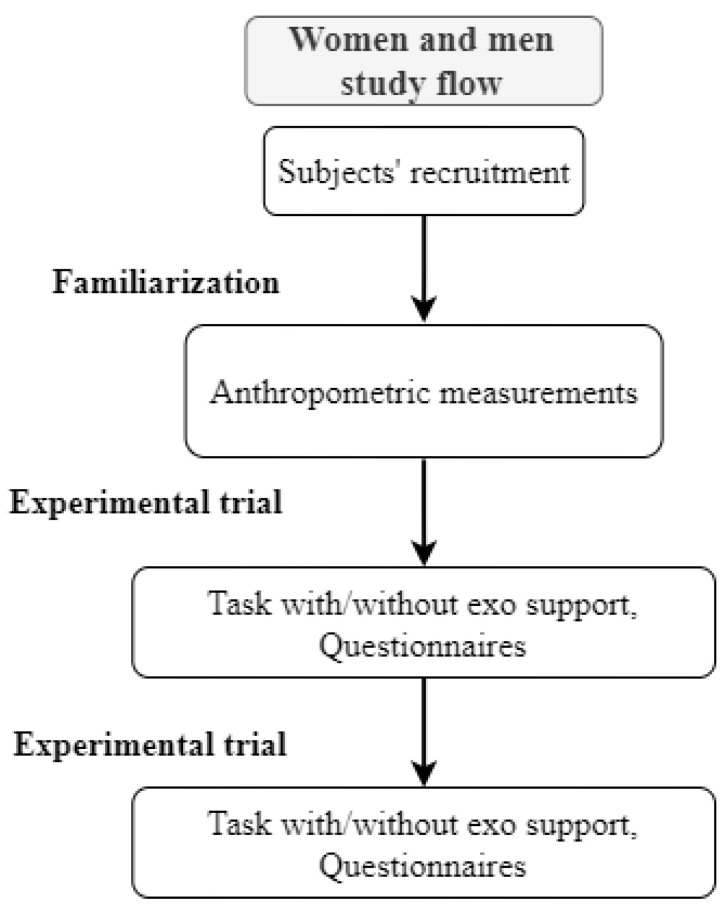
Study flow. Note: exo support = passive support.

**Figure 2 biomimetics-09-00601-f002:**
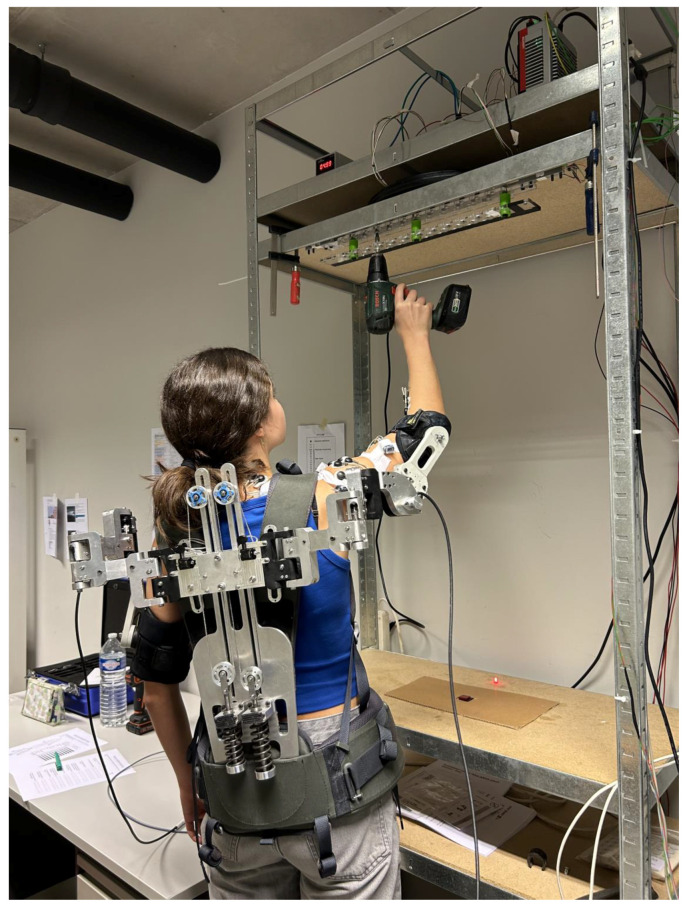
Laboratory setting with the execution of the industrial overhead task.

**Figure 3 biomimetics-09-00601-f003:**
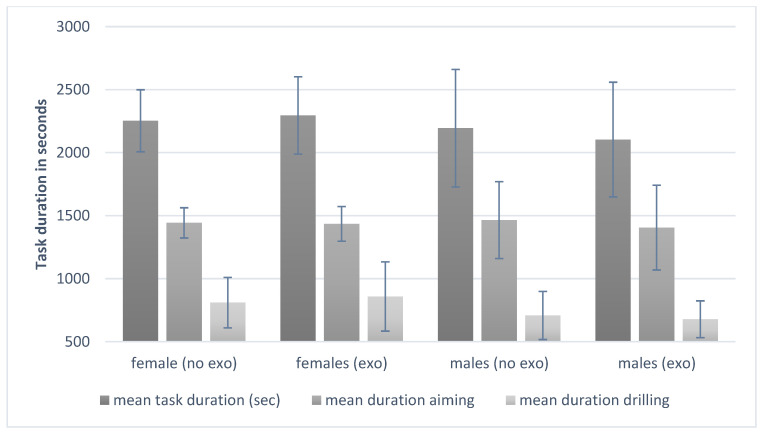
Results of the task duration for the different conditions.

**Figure 4 biomimetics-09-00601-f004:**
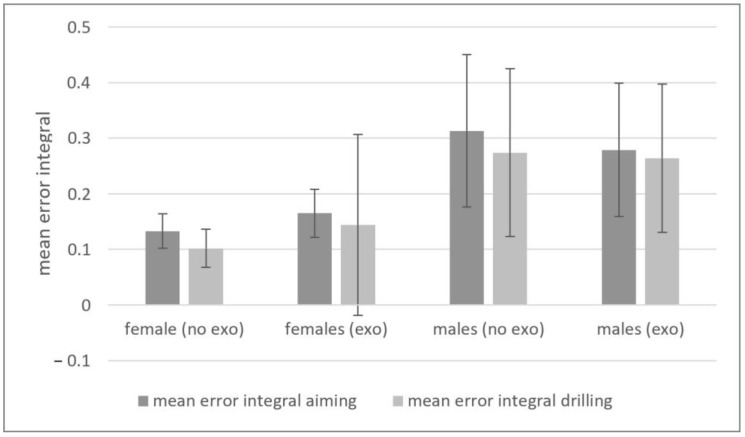
Results of the mean error integrals for the different conditions.

**Table 1 biomimetics-09-00601-t001:** Overview of the participant data.

Anthropometrics	Female n = 20 (MW ± SD)	Male n = 16 (MW ± SD)	F (1,34) *p*
Body height (m)	1.65 ± 0.04	1.81 ± 0.06	93.592 **<0.001**
Body mass (kg)	61.3 ± 8.5	81.4 ± 7.6	54.495 **<0.001**
BMI	22.5 ± 2.7	24.8 ± 2.4	7.042 **0.012**
Fingertip height (m)	2.09 ± 6.9	2.35 ± 8.3	99.773 **<0.001**
Fingertip height (m) with elbow at 90 degrees	1.73 ± 5.4	1.92 ± 4.1	126.463 **<0.001**

**Table 2 biomimetics-09-00601-t002:** Task performance data of both genders (with and without exoskeleton use).

Performance	Female noExo (M ± SD)	Female Exo (M ± SD)	Male noExo (M ± SD)	Male Exo (M ± SD)	Gender Differences F *p* eta^2^	Interactions no Exo/Exo and Gender F *p* eta^2^
mean task duration (s)	2253.7 ± 246	2295.1 ± 307.2	2194.9 ± 467.1	2104 ± 455.3	0.954 0.327 0.030	3.492 0.071 0.098
mean duration aiming	1443 ± 120.7	1435.5 ± 138.7	1465 ± 305.8	1405 ± 336.8	0.142 0.709 0.005	1.257 0.271 0.038
mean duration drilling	810.5 ± 200.8	859.6 ± 275.8	708.6 ± 191.2	678.5 ± 146.5	2.149 0.152 0.063	1.609 0.214 0.048
mean error integral aiming	0.133 ± 0.031	0.165± 0.043	0.313 ± 0.137	0.279 ± 0.120	**25.153** **<0.001** **0.440**	**5.509** **0.025** **0.147**
mean error integral drilling	0.102 ± 0.034	0.144 ± 0.163	0.274 ± 0.151	0.264 ± 0.133	**16.766** **<0.001** **0.344**	1.717 0.199 0.051

Note: M = mean, SD = standard deviation.

## Data Availability

The data set can be requested via the corresponding author.

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
