# Peer review of "Gender Differences in Performing an Overhead Drilling Task Using an Exoskeleton—A Cross-Sectional Study"

_biomimetics, 2024, doi:10.3390/biomimetics9100601_

Round 1

Reviewer 1 Report

Comments and Suggestions for Authors

Abstract

L10: Type of strain and comfort referred to here is unclear.

L15: Please state "controlling for BMI" as "BMI was included as covariate".

Introduction

L39-40: Some studies have shown this already. A good example is:

Ojelade, A., Morris, W., Kim, S., Kelson, D., Srinivasan, D., Smets, M., & Nussbaum, M. A. (2023). Three passive arm-support exoskeletons have inconsistent effects on muscle activity, posture, and perceived exertion during diverse simulated pseudo-static overhead nutrunning tasks. Applied Ergonomics, 110, 104015.

Please include this citation to make your argument stronger.

Materials and Methods

L93: Obviously, more detail is needed! What are the IVs, how did you alternate the order of presentation?

L107 and L112: Why did you limit your study to only right-handed participants?

L114: Why did you instruct your participants to refrain from coffee?

L116-120: This part should be in the overall participants sections and since it does not contribute to "restrictions and prohibitions"

L126-127: This material can be presented in a more concise manner

L129: Several concerns with the drilling subsection. How did you determine a successful completed task? Why did you include force sensors and accelerometers? How did you determine an error? Performance here is still unclear.

L145: Why were the two different Borg scale used for males and females? Also, it is unclear what the authors are referring to as "verbal categorization".

Lastly, how did you familiarize the participants to this scale?

L175: I think "trial" here should be "experimental session"

L204: what is the independent variable for your one-way ANOVA

Results:

It will be interesting to see some of the results explained with graphs. This section is rather "dull" now with the tables.

L212:

Comments on the Quality of English Language

L177: The word "foreseen" is unclear in this context.

L212: "may be" should be "is"

L219: change "if" to "when". Also please check for this throughout.

Author Response

We thank the reviewers for the time they spent on carefully and critically reading our manuscript. We also thank you for the opportunity to revise the manuscript accordingly. We answered each comment of the reviews point by point and highlighted changes in the manuscript in yellow. We think that these edits improved the overall quality of our manuscript and hope we met the criteria for publication of our manuscript.

Abstract

  1. L10: Type of strain and comfort referred to here is unclear.

Answer:

We agree with the reviewers´ concern and changed the clarified the sentence as follows in line 12:

“This study aimed to compare the performance as well as subjective body strain and comfort of men and women using an upper-body exoskeleton.“

  1. L15: Please state "controlling for BMI" as "BMI was included as covariate".

Answer: We thank the reviewer for this suggestion and changed it accordingly (cf. line 17)

“ The body mass index (BMI) was included as a covariate.”

Introduction

  1. L39-40: Some studies have shown this already. A good example is:

    Ojelade, A., Morris, W., Kim, S., Kelson, D., Srinivasan, D., Smets, M., & Nussbaum, M. A. (2023). Three passive arm-support exoskeletons have inconsistent effects on muscle activity, posture, and perceived exertion during diverse simulated pseudo-static overhead nutrunning tasks. Applied Ergonomics, 110, 104015.

    Please include this citation to make your argument stronger.

Answer:

We thank the reviewer for this comment and added the reference accordingly in line 43.

Materials and Methods

  1. L93: Obviously, more detail is needed! What are the IVs, how did you alternate the order of presentation?

Answer:

We apologize if there was missing information. We agree with the reviewer and added the sentence in line 93ff:

“Since the exoskeleton was designed and constructed for male users, we hypothesize that women will perform worse during the drilling task, have a higher rating of perceived exertion and lower usability score when using the exoskeleton in comparison to male users”.

  1. L107 and L112: Why did you limit your study to only right-handed participants?

Answer:

Indeed, only right-handed participants were included to allow an overall standardized procedure. This was initially used as inclusion criteria because we were also interested in kinematics using IMU sensors. Unfortunately, there was interference between the sensors and the exoskeletons and no data could be retrieved. Moreover, the set up and it programming was built to use the right hand. For left-handed persons the set-up should have been mirrored to allow the same movement strategy for their preferred side. However, with respect to this aspect, we discussed that this would have led to additional confounding of the results, because we would have an additional covariate in this case which would have raised the sample and would have led to problems with the recruitment. Nevertheless, we agree with the reviewers´ concern and added this aspect to the limitation section as follows in line 386 ff.

“Next to this aspect, the experimental set-up was designed and restricted for right-handed persons. Therefore, the results do not represent potential differences of the performance of left-handed individuals or could be generalized for left-handed persons.”

  1. L114: Why did you instruct your participants to refrain from coffee?

Answer:We thank the reviewer for this comment. We know from previous studies that coffee contains caffeine which has been shown to improve cognitive and physical performance. To control for this confounding factor, we requested participants to refrain from coffee and caffeine beverages.

We specified the section in line 131 ff as follows:

“Subjects were asked not to drink caffeine-containing beverages or alcohol and to refrain from vigorous training 24 hours before testing to avoid confounding influences on cognitive and physical performance.”

  1. L116-120: This part should be in the overall participants sections and since it does not contribute to "restrictions and prohibitions"

Answer:

We thank the reviewer for this comment and transferred the section accordingly.

  1. L126-127: This material can be presented in a more concise manner

Answer:

According to the reviewers´ suggestion, we presented this section in more detail. The section is now in line 112ff and reads as follows:

“2.2.2. Inclusion and Exclusion Criteria

Biological females and males aged 18 and older, right-handed, and without a history or current manifestation of musculoskeletal disorders or injuries in the upper extremity within the past 12 months were included. Participants were excluded if they had any injuries or health-related complaints. Only right-handed participants were included.

2.2.3 Participant description

Participants were recruited on or in the surroundings of the VUB campus in two rounds. First a number of N=16 males were examined. Afterwards a total number of N = 21 female were incorporated to the experiment. N = 1 female needed to be excluded due to poor data.

The mean age of the sample was 31.2 ± 14.5 years for the females and 31.9 ± 9.1 year for the males.

The anthropometrics naturally differed between female and male participants with higher values for males (cf. Table 1). As also the BMI differed significantly, the performance data had to be controlled for BMI.

  1. L129: Several concerns with the drilling subsection. How did you determine a successful completed task? Why did you include force sensors and accelerometers? How did you determine an error? Performance here is still unclear.

Answer:

We agree with the reviewer that we should have better clarified the measures of drilling performance.

We determined that a task was successfully completed until all 20 bolts were tightened (mean task duration). We adapted text as we did not use accelerometers, but an RM22 Rotary Magnetic Encoder, which was integrated in the exoskeleton’s right shoulder joint to track the exoskeleton’s shoulder elevation angle (1kHz; RLS Slovenia).

During the overhead drilling task, the exoskeleton’s encoder data and load cell data of the sensorized overhead work bench were acquired at a frequency of 1000 Hz, using Beckhoff components (Beckhoff, Germany) interconnected via an Ethercat communication protocol. Pressing the push button at hip-level indicated movement initiation. The load cells connected to the bolts allowed tracking when the screwdriver bit was touching a bolt. The encoder data were used to differentiate between lifting the arm from hip level towards the bolts, and the slower aiming process before the bolts were reached. These data allowed to segment all acquired data into a lifting and aiming phase, and operative phase. In the latter phase, the bolts were tightened. Data from the load cells connected to the plexiglass surrounding the bolts documented the error. In order to encompass the duration of the error, the error score was computed as the time-integrated output from the force sensors connected to the plexiglass plates.

We therefore completely revised this section: It now reads as follows starting in line 142:

To quantify the working performance and facilitate segmentation of the acquired signals, force sensors and accelerometers are attached to the overhead working setup and the weights that will be lifted. To determine the proper overhead height, the method of Sood et al. [25] was used (hand height with the shoulder and elbow in a 90-degree angle + 0.4 x (hand height with the arm in full extension - hand height with the shoulder and elbow in a 90-degree angle). To evaluate the precision performance in overhead work, a custom task with high to moderate test-retest reliability (adapted from Kim et al. [26]) was developed. In this task, participants used an electric screwdriver (Black & Decker, US, 1.14 kg) to tighten 20 pre-inserted bolts in an aluminum plate positioned at overhead height. We determined that a task was successfully completed until all 20 bolts were tightened (mean task duration). The bolts were partially screwed into the plate using self-fixing nuts, standardizing the distance between the bolt heads and the plate. A plexiglass mask was placed over the bolts to interfere with the screwdriver tip if the screwing trajectory deviated by more than ±2° from perpendicular to the plate [26]. Participants pressed a push button at pelvic crest height, tightened a bolt at overhead height, and pressed the button again (cf. Fig. 2), repeating this process for all bolts. They were instructed to perform the task quickly and accurately, minimizing errors. Sensorized aluminum and plexiglass plates tracked contact between the screwdriver bit and the bolt, as well as screwing errors. Pressing the push button marked the start of each movement.

During the overhead drilling task, the exoskeleton’s encoder data and load cell data of the sensorized overhead work bench were acquired at a frequency of 1000 Hz, using Beckhoff components (Beckhoff, Germany) interconnected via an Ethercat communication protocol. Pressing the push button at hip-level indicated movement initiation. The load cells connected to the bolts allowed tracking when the screwdriver bit was touching a bolt. The encoder data were used to differentiate between lifting the arm from hip level towards the bolts, and the slower aiming process before the bolts were reached. These data allowed to segment all acquired data into a lifting and aiming phase, and operative phase. In the latter phase, the bolts were tightened. Data from the load cells connected to the plexiglass surrounding the bolts documented the error. In order to encompass the duration of the error, the error score was computed as the time-integrated output from the force sensors connected to the plexiglass plates.

The Exo4Work's right shoulder joint, fitted with an RM22 Rotary Magnetic Encoder, tracked shoulder elevation angle (1 kHz; RLS, Slovenia). All measurements were recorded on the right side, so tasks were performed right-handed, with the left hand at rest.

  1. L145: Why were the two different Borg scale used for males and females? Also, it is unclear what the authors are referring to as "verbal categorization". Lastly, how did you familiarize the participants to this scale?

Answer:

We agree with the reviewers´ concern. We planned the experiment in two different waves, as the male part was one part of a PhD project and the gender comparison part of a different approach. Indeed, we used 2 different Borg scales, which was an error. We applied the 100 CR scale, since this one is more precise in expressing perceived exertion. To compare the scales, we downscaled the female 100 CR scale to the 10-point scale and the participants were instructed express the RPE verbally. The instruction was also part of the familarization process.

We changed the section as follows in line 178:

For the male participants in the study, the 10-point Borg scale was used [27, 28].

For the second part of the study, including female participants, the subjective assessment of the perceived exertion was carried out using the 100-point Borg scale [27-30], which is more precise in accounting for the RPE. This scale allows participants to subjectively assess the intensity of exertion after each drilling trial, with 0 representing no exertion and 100 indicating maximal effort. The assessment refers to the overall perceived exertion and specific body regions, such as the neck, shoulders, arms, upper and lower back, and legs. A chart was shown for this purpose. The rating of the perceived exertion was expressed verbally by the participants.

The familiarization to the RPE scales were done by verbal instructions in the initial session. Familiarizing participants with the scale involved presenting the scale and asking for their ratings during the experimental trial. This process helped participants become accustomed to the task and measurement tools being used.

To compare both CR scales, we downscaled the female 100 CR scale to the 10-point scale.

  1. L175: I think "trial" here should be "experimental session"

Answer:

We apologize if the section was misleading. The rating was done after each drilling trial. We specified this accordingly in line 182.

  1. L204: what is the independent variable for your one-way ANOVA

Answer:

We added the independent variable to the text. It now reads as follows in line 245:

To analyze the data, one-way ANOVA was performed to compare demographic variables (body weight and height, BMI, fingertip height, and fingertip height with elbow at 90 degrees) to detect gender differences.

  1. Results:

It will be interesting to see some of the results explained with graphs. This section is rather "dull" now with the tables. L212:

Answer:

We thank the reviewer for this suggestion and added Figure 3 (line 260) and Figure 4 (line 270) to visualize the results.

Comments on the Quality of English Language

L177: The word "foreseen" is unclear in this context.

L212: "may be" should be "is"

Answer:

We thank the reviewer for this comment and changed it accordingly.

L219: change "if" to "when". Also please check for this throughout.

Answer:

We thank the reviewer for this advice. We did a comprehensive language editing for the whole manuscript.

Reviewer 2 Report

Comments and Suggestions for Authors

Line 36/37: "range of motion" It seems unlikely that there are significant differences in the anatomical range of motion of human joints between the genders. An explanation of what is meant by this is therefore required. 

Line 37/397: "[5]" I can't find this publication! Please provide more Informations about it!

Line 47-49: This statement is about using back-supporting exoskeletons. It would be good to check if this also applies to shoulder-supporting exoskeletons. If it doesn't, you can leave this part out. 

Line 50/416: Please note the above explanations. Paper #13 This paper does not appear to address the issue of gender differences in range of motion. Unfortunately, paper no. [14] cannot be found under the following link: https://journals.lww.com/ajpmr/Pages/issuelist.aspx?year=2015.

This paper does not seem to exist!

Line 103-104: Please check the spelling here! 

Line 107-110: One inclusion criterion for men is that they must be industrial workers. This criterion is not applied to women, which raises the question of why this is the case. This could be a limitation of this study.

Line 137: What were the reasons for the use of two distinct Borg scales? It appears that two distinct and independent studies were conducted.

Line 155: Please clarify whether you adjusted the pretension of the spring on an individual basis for each subject. Additionally, please indicate whether you anticipate that the amount of support will influence the execution of the specific task.

Line 156-157: The maximum support torque of 3 Nm appears to be relatively low, particularly in view of the hysteresis curve of a passive exoskeleton, whereby less support is returned than is introduced into the system. Please provide an explanation of the actual efficiency of the exoskeleton, taking hysteresis into account.

Line 196: Is it reasonable to assume that the angle of the shoulder joint correlates with the angle of the exoskeleton?

Line 244-247: Please provide at least one relevant reference to support these general statements.

Line 271-272: see my comments above

Line 329: Please mention the limitation that the shoulder angle is NOT the same as the exoskeleton angle. You have not measured the shoulder angle.

Line 339: In line 110 you write that the inclusion criterion for men is industrial workers. This does not fit with this statement!

Please respond to the following assumption: It appears that two distinct and independent studies were conducted. See line 137ff, 143ff and 339ff

Comments on the Quality of English Language

Line 103-104: Please check the spelling here! 

Line 228: "males" not mals

Author Response

We thank the reviewers for the time they spent on carefully and critically reading our manuscript. We also thank you for the opportunity to revise the manuscript accordingly. We answered each comment of the reviews point by point and highlighted changes in the manuscript in yellow. We think that these edits improved the overall quality of our manuscript and hope we met the criteria for publication of our manuscript.

  1. Line 36/37: "range of motion" It seems unlikely that there are significant differences in the anatomical range of motion of human joints between the genders. An explanation of what is meant by this is therefore required. 

Answer:

We apologize if this aspect was misleading and clarified this aspect. It now reads as follows starting in line 37:

Men and women often exhibit physiological and biomechanical differences, which can influence the performance and effectiveness of assistive devices like exoskeletons [4]. Factors such as muscle strength or range of motion according to body dimensions vary between genders and may lead to differing interactions with the exoskeleton [5]

Line 37/397: "[5]" I can't find this publication! Please provide more Informations about it!

Answer:

We apologize for the confusion with the references. There was a mistake in our citation program and the reference list was mixed up. Of course, it is the task of the whole author team to double check on this, before uploading the final manuscript. The full reference is:

Leibman, D., & Choi, H. (2024, August). Exoskeleton Use During Lifting Tasks May Impair Physical and Cognitive Performances among Novice Users. In Proceedings of the Human Factors and Ergonomics Society Annual Meeting (p. 10711813241264209). Sage CA: Los Angeles, CA: SAGE Publications.

  1. Line 47-49: This statement is about using back-supporting exoskeletons. It would be good to check if this also applies to shoulder-supporting exoskeletons. If it doesn't, you can leave this part out. 

Answer:

The thank the reviewer for this comment. However, we think, that independent of support area of the exoskeleton (shoulder or back), the exoskeleton is worn on the upper body and the described mechanisms can also be true for the shoulder exoskeletons. However, we specified the section as follows (line 50 ff):

These differences may influence how body weight and the additional weight of the exoskeleton are distributed and how the exoskeleton stabilizes the body during use. In addition, women might adopt different postural adjustments due to their wider pelvis and different center of gravity, which can further impact the exoskeleton’s fit and function [13].

  1. Line 50/416: Please note the above explanations. Paper #13 This paper does not appear to address the issue of gender differences in range of motion. Unfortunately, paper no. [14] cannot be found under the following link: https://journals.lww.com/ajpmr/Pages/issuelist.aspx?year=2015.

This paper does not seem to exist!

Answer:

We agree with the reviewer. As already reported, there was a mistake in our reference program. We now corrected the whole reference section.

  1. Line 103-104: Please check the spelling here! 

Answer:

We thank the reviewer for this comment and changed the spelling as follows starting in line 108 ff:

To compare male and female participants without and without exoskeleton support, a power calculation with G*Power revealed a total sample size of 41 subjects (ANOVA; Fixed effects, main effects and interactions, ß-Power of 0.8, estimated effect size of 0.45 and alpha of 0.05).

  1. Line 107-110: One inclusion criterion for men is that they must be industrial workers. This criterion is not applied to women, which raises the question of why this is the case. This could be a limitation of this study.

Answer:

We apologize that our description was misleading. The experiment was conducted to mimic an industrial setting and to transfer the setting to real-world scenarios at a certainn stage. This was not done yet and therefore we deleted this aspect from the description.

  1. Line 137: What were the reasons for the use of two distinct Borg scales? It appears that two distinct and independent studies were conducted.

Answer:

We agree with the reviewers´ concern. The study was conducted in two steps. We planned the experiment in two different waves, as the male part was one part of a PhD project and the gender comparison part of a different approach. Indeed, we used 2 different Borg scales, which was an error. We applied the 100 CR scale, since this one is more precise in expressing perceived exertion. To compare the scales, we downscaled the female 100 CR scale to the 10-point scale and the participants were instructed express the RPE verbally. The instruction was also part of the familarization process. Familiarizing participants with the scale involved presenting the scale and asking for their ratings during the experimental trial. This process helps participants become accustomed to the task and measurement tools being used.

To compare both CR scales, we downscaled the female 100 CR scale to the 10-point scale.

We changed the section as follows in line 178 ff:

For the male participants in the study, the 10-point Borg scale was used [27, 28].

For the second part of the study, including female participants, the subjective assessment of the perceived exertion was carried out using the 100-point Borg scale [27-30], which is more precise in accounting for the RPE. This scale allows participants to subjectively assess the intensity of exertion after each drilling trial, with 0 representing no exertion and 100 indicating maximal effort. The assessment refers to the overall perceived exertion and specific body regions, such as the neck, shoulders, arms, upper and lower back, and legs. A chart was shown for this purpose. The rating of the perceived exertion was expressed verbally by the participants.

The familiarization to the RPE scales were done by verbal instructions in the initial session. Familiarizing participants with the scale involved presenting the scale and asking for their ratings during the experimental trial. This process helped participants become accustomed to the task and measurement tools being used.

To compare both CR scales, we downscaled the female 100 CR scale to the 10-point scale.

  1. Line 155: Please clarify whether you adjusted the pretension of the spring on an individual basis for each subject. Additionally, please indicate whether you anticipate that the amount of support will influence the execution of the specific task.

Answer:

We agree with the reviewers´ concern. The adjustment of the pretension of the spring was standardized and not individualized for every subject. We clarified this in the text as follows starting in line 206:

The set-up including pretension of the spring was standardized and being kept the same for each subject. The exoskeleton can be worn on the body like a backpack with a hip belt, that ensures the transfer of the weight from the back to the legs. The device was attached to the upper arms with Velcro straps, similar to commercially available devices. For more information about the exoskeleton (kinematics, torque distribution and hysteresis readers are referred to Rossini et al., 2021[32].  

  1. Line 156-157: The maximum support torque of 3 Nm appears to be relatively low, particularly in view of the hysteresis curve of a passive exoskeleton, whereby less support is returned than is introduced into the system. Please provide an explanation of the actual efficiency of the exoskeleton, taking hysteresis into account.

Answer:

We thank the reviewer for this comment. The support is capable of assisting the user with movements above shoulder height, and 3 Nm torque allows the arm to be lowered without adding resistance, which would otherwise increase metabolic cost, reduce work efficiency and increase discomfort. The description of the initial development and testing of the exoskeleton regarding kinematics, torque distribution and hysteresis can be found here:

Reference: Rossini M, De Bock S, van der Have A, Flynn L, Rodriguez-Cianca D, De Pauw K, Lefeber D, Geeroms J, Rodriguez-Guerrero C. Design and evaluation of a passive cable-driven occupational shoulder exoskeleton. IEEE Transactions on Medical Robotics and Bionics 2021; 3(4): 1020-1031.

We therefore added the following sentence in line 210:

For more information about the exoskeleton (kinematics, torque distribution and hysteresis readers are referred to Rossini et al., 2021[32].  

  1. Line 196: Is it reasonable to assume that the angle of the shoulder joint correlates with the angle of the exoskeleton?

Answer:

We agree with the reviewer, that this would have been interesting to evaluate this aspect. However, we are not able to determine this aspect since we have not attached IMU sensors to the participants. 

  1. Line 244-247: Please provide at least one relevant reference to support these general statements.

Answer:

We provided references there.

  1. Line 271-272: see my comments above

Answer:

We provided references there.

  1. Line 329: Please mention the limitation that the shoulder angle is NOT the same as the exoskeleton angle. You have not measured the shoulder angle.

Answer:

We thank the reviewer for this comment and added this into the limitation section as follows in line 389 ff:

We have to address that the individuals´ shoulder angle might differ from the exoskeletons shoulder support angle. We only measured the angle of the exoskeleton shoulder joint.

  1. Line 339: In line 110 you write that the inclusion criterion for men is industrial workers. This does not fit with this statement!

Answer:

We apologize if the method section was misleading. We corrected the method section accordingly.

  1. Please respond to the following assumption: It appears that two distinct and independent studies were conducted. See line 137ff, 143ff and 339ff

Answer:

Indeed, this study contains two parts, in which the first part of the experiment included male participants and the second part of the experiment included female participants.

We clarified this in the method section as follows starting in line 120:

Participants were recruited on or in the surroundings of the VUB campus in two rounds. First a number of N=16 males were examined. Afterwards a total number of N = 21 female were incorporated to the experiment. N = 1 female needed to be excluded due to poor data quality.

Comments on the Quality of English Language

  1. Line 103-104: Please check the spelling here! 

Answer:

  1. We thank the reviewer and corrected the spelling (cf. comment 5.)

Line 228: "males" not mals

Answer:

We thank the reviewer for this advice and changed it accordingly.

Round 2

Reviewer 2 Report

Comments and Suggestions for Authors

Line 210: Please check the spelling, there might be an "(" too much.

Author Response

Dear reviewer, we thank you again for carefully reading our manuscript and your input.

We double checked your comment:

Line 210: Please check the spelling, there might be an "(" too much.

We had a missing bracket ")" because we wanted to give the examples in brackets. We now have added this in line 210 accordingly.